# Morphology Optimization of Residential Communities towards Maximizing Energy Self-Sufficiency in the Hot Summer Cold Winter Climate Zone of China

Yuan Zhou [1,2,3], Hongcheng Liu [1,2,3], Xing Xiong [1,2,3] and Xiaojun Li [1,2,3,*]

1   School of Architecture and Planning, Hunan University, Changsha 410082, China;
    yzhou123@hnu.edu.cn (Y.Z.); lhc913@hnu.edu.cn (H.L.); xing0726@hnu.edu.cn (X.X.)
2   Hunan Key Laboratory of Sciences of Urban and Rural Human Settlements in Hilly Areas, Hunan University,
    Changsha 410082, China
3   Hunan International Innovation Cooperation Base on Science and Technology of Local Architecture,
    Changsha 410082, China
*   Correspondence: junglix@hnu.edu.cn

**Abstract:** Further research is needed on the capability of residential communities to achieve energy self-sufficiency under the constraints of current standards of land use, in particular for the Hot Summer and Cold Winter climate zone (HSCW) of China, where the majority of communities are dominated by high floor-area ratios, thus high-rise dwellings, namely less solar potential per unit floor area, while most residents adopt a "part-time, part-space" pattern of intermittent energy use behavior, thus using relatively low energy per unit floor area. This study examines 150 communities in Changsha to identify morphological indicators and develop a prototype model utilizing the Grasshopper platform. Community morphology is simulated and optimized by taking building location, orientation, and number of floors as independent variables and building energy consumption, solar PV generation, and energy self-sufficiency rate as dependent variables. The results reveal that the morphology optimization can achieve a 4.26% decrease in building energy consumption, a 45% increase in PV generation, and a 13.2% enhancement in energy self-sufficiency, with the optimal being 39%. It highlights that energy self-sufficiency cannot be achieved solely through morphology improvements. Moreover, the study underscores the crucial role of community orientation in maximizing energy self-sufficiency, with the south–north orientation identified as the most beneficial. Additionally, a layout characterized by a horizontally closed and staggered pattern and a vertically scattered arrangement emerges as favorable for enhancing energy self-sufficiency. These findings underscore the importance of considering morphological factors, particularly community orientation, in striving towards energy-self-sufficient high-rise residential communities within the HSCW climate zone of China.

**Keywords:** community morphology; parameter design; energy self-sufficiency; solar PV generation; zero energy community

## 1. Introduction

### 1.1. Background

In response to the challenges posed by climate change and energy insecurity, countries worldwide have successively set dual carbon targets tailored to their unique national circumstances. The building sector, known for its significant share of energy consumption and carbon emissions, stands out as a critical area with immense energy and carbon reduction potential. Following the improvement of building energy efficiency since the energy crisis in the 1970s, renewable energy, such as photovoltaic, has been popularized. Solar energy storage has been demonstrated to achieve near-zero energy buildings in the Tibetan plateau [1]. Similarly, solar energy utilization is highly relevant to the zero-energy building

design process [2] and has further been demonstrated in energy-positive buildings [3]. As a result, energy self-sufficiency is receiving increasing attention, and the integration of distributed photovoltaics and buildings has shifted from a building scale to an urban scale. Emerging research increasingly focuses on energy self-sufficiency at the community level, resulting in various models such as renewable energy-sharing communities, zero energy communities, and plus energy communities, primarily in Europe and the US. Noteworthy studies, like Nematchoua's research [4], have outlined how extensive renovations at the neighborhood scale can achieve the 'nearly zero-energy' objective. Similarly, Shandiz et al. [5] proposed a comprehensive energy master planning framework to facilitate the design and development of net-zero emission communities. Concurrently, European municipalities are actively pursuing integrating cross-sectoral policies to realize the common objective of establishing zero-energy communities [6]. However, due to differences in geography, climate, and building regulations, the balance of energy demand and renewable supply achievable for communities in Europe and the US may not work in China, in particular because most new communities in China are dominated by the high floor area ratio, which is used to describe the ratio of a building's total floor area to the size of the land on which it is situated, as a result of high-rise buildings.

Energy self-sufficiency is typically obtained by reducing energy consumption and increasing on-site energy production [7]. Researchers have found that prioritizing generating and storing electricity is crucial for the design of energy-self-sufficient communities [8]. Factors influencing energy self-sufficiency rate include building envelope performance, morphological factors, service systems, renewable supply, occupant behavior, and operation management. It has been demonstrated that materials with low thermal conductivity can limit building energy storage efficiency [9]. Critical factors for improving energy efficiency in tropical buildings include insulation, glazing properties, and the window-to-wall ratio [10]. According to Kadrić et al. [11], upgrading external walls and improving heating system efficiency could effectively reduce energy consumption. Aneli et al. [12] have developed an innovative approach, demonstrating how typical residential buildings achieve energy self-sufficiency by utilizing solar and wind energy sources. Moreover, occupant behavior shows its significance for sustainable building energy performance [13]. Among all, morphology plays an important role, not only affecting energy consumption but also serving as a prerequisite for other factors, such as solar energy potential. It is found that urban morphology parameters can be linked to energy performance indicators [14], and optimizing urban forms can minimize building energy consumption and maximize solar energy potential, with varying influences [15–17].

*1.2. Related Work and Research Gaps*

In order to study the correlation between community morphology and building energy consumption as well as solar PV potential, a literature review is carried out, with the findings summarized in Table A1. Although the reviewed research objects of neighborhood, community, and residence may vary in size, they are all comprised of a group of dwellings and therefore are relevant for this research. As shown in Table A1, morphological factors that influence building energy consumption include the number of floors [18], building (urban) density [19], layout of the building group [20], building type, floor area ratio, standard deviation of building heights [21], shape factor [22], building (neighborhood) orientation [23], open space rate [24], and building spacing coefficients [25], which may vary among climates. And dynamic thermal simulation tools such as Virvil Sketchup, UMI, Grasshopper (Ladybug), Citysim, and Design Builder are employed to explore the relationship between morphological factors and building energy consumption. Among all, Grasshopper emerged as a comprehensive and robust parameter design and optimization tool in urban-scale modeling. Studies have also explored how the spatial morphology of a building group influences its solar PV potential. For instance, Mohajeri [26] demonstrated that compatness is a key urban form parameter that affects the accessibility of solar energy. Zhang et al. [27] compared PV potential across various neighborhood types and

determined that the courtyard and hybrid neighborhoods exhibit double the solar energy utilization potential of the tower and slab configurations. Additionally, Lobaccaro et al. [28] highlighted that optimizing morphological factors such as building orientation, height, and spacing could boost solar energy utilization potential by up to 25%. Moreover, site layout of the neighborhood, associated position of buildings [29], floor area ratio, building average spacing [30], and urban form parameters [31] may also affect solar energy utilization. As with building energy consumption, the major morphological factors affecting solar PV potential vary among weather conditions, with software tools like ArcGIS and Radiance commonly utilized for analysis. Generally, the morphological factors affecting building energy consumption and solar PV potential can be categorized into urban scale, community scale, and building scale; thus, the morphological factors adaptable at the community scale can be obtained, including building type, number of floors, building and community orientations, building layout, building density, floor area ratio, and community dimensions, as shown in Figure A1.

Research has shown that the morphology of a community has a significant influence on its building energy consumption and solar PV potential, and the influence may vary greatly among factors and climates. However, most research focuses on the impacts of community morphology on building energy consumption and solar PV potential separately, with few on the balance between energy consumption and solar PV generation. Moreover, in order to measure the spatial relationship of a community, morphological factors are quantified to derive parameters such as canyon aspect ratio (width to depth or height ratio of a canyon), street aspect ratio (width to length ratio of a street), and space open rate (percentage of available or vacant space in a given area or property). Although these parameters can reflect the relative position relationship between buildings to a certain extent, they do not encompass all possibilities of layouts within a site and therefore possess limitations for the optimization of community morphology.

### 1.3. Objectives

As stated above, the energy self-sufficiency potential of a PV-integrated community is determined by comparing energy use and on-site PV generation within a given time, both of which can be impacted by community morphology. The Hot Summer and Cold Winter (HSCW) climate zone in China, characterized by sweltering summers and cold winters, is inhabited by around 40% of the country's population. Residents in this area have adopted a "part-time, part-space heating and cooling" behavior pattern to cope with the specific weather, resulting in lower energy consumption than a "full-time, full-space" mode. Given the high population density in Chinese cities, new residential developments are dominated by high floor-area ratios, resulting in high-rise buildings, as in the HSCW zone. Even though residents in this zone consume less energy, the potential for solar PV generation per unit floor area is also limited in high-rise residential buildings. As new residential communities continue to expand and develop in the area, there is an opportunity for early-stage morphology improvements and the integration of low-carbon technologies like solar PV on all available opaque building surfaces. Besides being driven by building energy efficiency policies and regulations, the energy consumption of new residential developments has decreased significantly, placing greater emphasis on on-site PV generation capacities, which are technically and economically viable for distributed application at both building and community scales. This trend has been globally endorsed and promoted. To maximize the energy self-sufficiency potential of a PV-integrated community, it is essential to study the morphology improvements of residential developments in the HSCW zone. Moreover, it can serve as a good reference for developing and adjusting land use regulations and standards that govern the region's design and planning of residential developments.

This study aims to explore the morphology optimization of high-rise residential developments in the China's Hot Summer and Cold Winter climate zone towards achieving maximum energy self-sufficiency. The primary objective is to contribute to the design and

planning of zero-energy communities, explicitly focusing on residential developments in Changsha, a representative HSCW city in China. It seeks to address the following inquiries:

(1) Can energy self-sufficiency be attained through PV-integrated residential developments in the area?

(2) What are the critical morphological factors influencing energy self-sufficiency, and how can the energy self-sufficiency rate be maximized?

The paper is organized as follows: Section 2 introduces materials and methods, including establishing typical community and building models, parameter design, and automatic optimization; Section 3 presents and analyzes the simulation results, including significance analysis, fitting analysis, and optimization strategy analysis; and Section 4 further discusses the results and limitations.

## 2. Materials and Methods

The study establishes typical community and building models based on a survey of residential communities and buildings in Changsha before carrying out parameter design and automatic optimization with the community morphology in Ladybug Tools. The research samples consist of communities built from 2018 to 2022 in various administrative districts of Changsha, including Furong, Tianxin, Yuelu, Kaifu, Yuhua, Wangcheng, Liuyang, Ningxiang, and Changsha County. In terms of community morphological factors (Figure A1), community size, building type, floor area ratio, and building density are kept constant in the investigation. Community size and building type data are collected from surveys, while floor area ratio and building density adhere to specified thresholds in regulations to maximize developers' interests. The optimization objectives encompass maximizing energy self-sufficiency, reducing building energy consumption, and increasing solar PV generation. The research proceeds in three steps:

Step 1: establishing the typical community model based on a morphology survey.

Step 2: establishing and validating the typical building model through the survey and literature review.

Step 3: conducting parameter design coupled with automatic optimization to derive optimal schemes, then exploring the correlation between the morphological factors and the objective values, respectively, and identifying the corresponding optimization strategies at the end.

### 2.1. The Community Model

A statistical analysis was conducted on 150 randomly selected communities in Changsha to identify typical characteristics related to community layout, building type, number of floors, and land area. The community layouts were classified into four main types: row (a specific layout where multiple dwellings are arranged in rows, characterized by a particular orientation and spacing), staggered row, courtyard, and hybrid (row + staggered row, row + courtyard, staggered row + courtyard). Among these, the row layout was the most prevalent at 69%, with courtyard, staggered row, and hybrid layouts being less common (Figure A2). Moving on to building types, slab dwellings dominated at 66%, followed by tower-slab dwellings at 17%, tower dwellings at 1%, and a mix of tower-slab with slab or tower dwellings at 7%, thus it can be concluded that it is dominated by slab dwelling communities (Figure A2). The statistical analysis of the average number of floors, based on the classification of the average number of floors for residential communities according to the Project Code for Residential Building, revealed that slab dwellings with 10–18 floors constituted the largest proportion (Figure A3). Furthermore, an analysis of community land areas categorized into six intervals showed that communities with a land area of 30,000–60,000 m$^2$ were the most common, representing 30.6% of the total sample (Figure A3).

The characteristics of the typical community are derived based on the above parameters, such as a land area ranging from 30,000 m$^2$ to 60,000 m$^2$, a row layout (Figure A4), and slab dwellings. Following the Urban Residential Area Planning and Design Standards, key

control indicators for the typical community are established based on sample data. These indicators include an average number of floors of 23 and a floor area ratio of 2.96. Then, a 197 m × 197 m square plot is established as the ideal site, with the orientation defaulted to be north–south, and the building control line is set 10 m inward from the site boundary line (Figure 1). The design parameters of the typical community are summarized in Table 1.

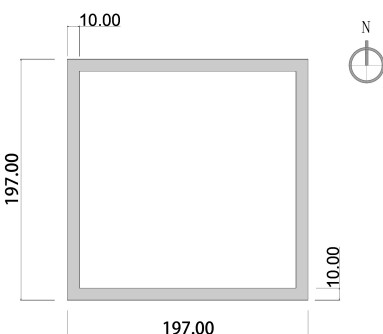

**Figure 1.** Plan prototype of the community.

**Table 1.** Design parameters of the typical community.

| Design Parameters of Community | Parameter Value |
| --- | --- |
| Land area | 38,800 (m$^2$) |
| Total floor area of buildings | 114,954 (m$^2$) |
| Floor area ratio | 2.96 |
| Building density | 12.9 (%) |
| Layer height | 3 (m) |
| The average number of floors | 23 (floors) |
| Number of buildings | 7 (buildings) |

*2.2. The Building Model*

According to the floor plan database for residential buildings constructed in Changsha from 2018 to 2022, the typical plan of the slab dwellings, the most dominant type in the area, is extracted (Figure 2), and a typical building model is established based on this prevalent design subsequently (Table A2). The parameter settings for the building envelope and HVAC system are determined by the guidelines outlined in DBJ43/T025-2022, "Design Standard for Energy Efficiency of Residential Buildings in Hunan Province" [32], and GB 55015-2021, "General Code for Energy Efficiency and Renewable Energy Application in Buildings" [33]. A fundamental assumption in the analysis is the utilization of air conditioning for both heating and cooling. In terms of solar PV, the polycrystalline silicon solar panel, the most popular PV type, is selected for the analysis. It has a photovoltaic conversion efficiency ranging from 14% to 19% [34], a photoelectric conversion factor of 17%, and a DC-to-AC inverter efficiency of 85%. In addition, it is assumed that both the roofs and opaque façades of the dwellings can be covered with solar PV. The allowable area ratio of the building envelope covered with PV panels is capped at 70%, accounting for deductions due to window area (30% of the façade) and unusable roof space. Moreover, according to Compagnon, specific thresholds for electricity generation per unit area should be set at 800 kWh/m$^2$/y for solar PV systems integrated on façades and 1000 kWh/m$^2$/y for those on roofs [35]. Considering the continuous reduction of costs and the gradual maturation of PV technology in the past years, the threshold of power generation in this study is set to be 800 kwh/m$^2$/y. The weather data utilized for simulation purposes was sourced from the SWERA database provided by EPWMAP (Mar 2015) meteorological data via Ladybug (1.4.0). In the context of Chinese architectural practices, residential buildings on the peripheries of communities often incorporate podiums to accommodate community-oriented commercial spaces. However, due to the distinct energy consumption patterns between residential and commercial buildings and the minimal impact of podiums on

high-rise residential structures' energy consumption and PV potentials, only dwellings without podiums are considered in this study.

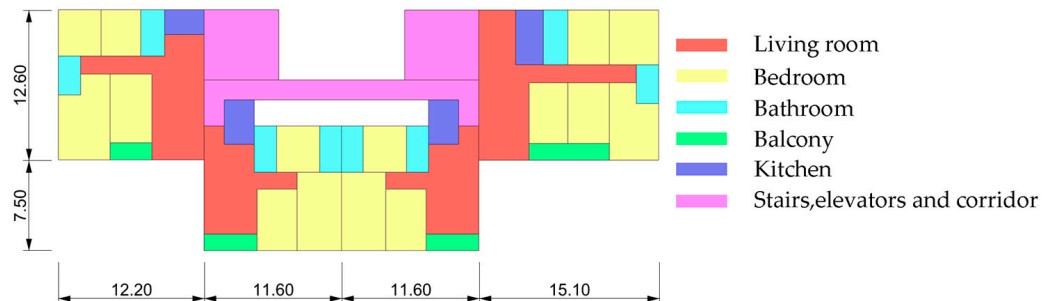

**Figure 2.** Typical residence floor plan.

Based on the parameter settings in Table A2, the dwelling's energy consumption and PV generation have been simulated and compared with current standards and existing literature studies to validate the model. The simulation results, detailed in Table 2, reveal that the annual electricity consumption for heating and cooling is 20.8 kWh/m$^2$. At the same time, the power generated by solar PV per unit area of valid building facades and roofs where PV can be laid is 97.64 kWh/m$^2$. It is important to note that the dwelling's air conditioner operation is based on a part-time and full-space mode to simplify the community model and expedite simulation. Thus, the obtained simulation results may be higher than those under the actual operation mode of "part-time and part-space" in the area. Notably, a key objective of one of China's National Key Research and Development Programs is to limit the future annual heating and cooling energy consumption of urban communities in the Hot Summer and Cold Winter zone to less than 20 kWh/m$^2$ [36]. Correspondingly, the reference value for heating and air conditioning energy consumption in the design of new residential buildings in Changsha is 13.7 kWh/m$^2$, as stated in the *Design standard for energy efficiency of residential buildings in the hot summer and cold winter zones (partial revision of the provisions for exposure of opinions)* [37]. While studies on current energy consumption patterns in Changsha dwellings reveal that annual heating and cooling energy consumption per unit area ranges between 27.7 kWh/m$^2$ and 40 kWh/m$^2$ [38–40]. Nevertheless, the simulation results from the study fall within an acceptable range. In terms of electricity generated by solar PV, studies indicate that the electricity yield per unit paved area varies from 35.6 kWh [41] to 143.07 kWh [42] in typical Hot Summer and Cold Winter cities like Changsha and Hangzhou. As above, the results of the study are just within the above range, further attesting to the model's accuracy. Verifying the typical building model can provide a reliable basis for the subsequent work of community morphology optimization.

**Table 2.** Simulation results of the typical building.

| Morphology of Model | Simulation Index | Index Value |
|---|---|---|
| | Average duration of daylight (h) | 7.75 |
| | Annual cooling energy consumption (kWh/m$^2$) | 9.79 |
| | Annual heating energy consumption (kWh/m$^2$) | 11.01 |
| | Annual heating and cooling energy consumption (kWh/m$^2$) | 20.8 |
| | Electricity-generating area of the building facades and roofs (m$^2$) | 2312.03 |
| | Electricity generated by solar PV per unit area of the valid building facades and roofs (kWh/m$^2$) | 97.64 |
| | Annual solar radiation of the valid area of building facades and roofs (kWh) | 225,750.5 (kWh) |

*2.3. Parametric Design and Optimization*

2.3.1. The Platform

Ladybug Tools, a simulation software based on Rhino and Grasshopper, surpasses other tools in dynamic simulation and analysis of energy performance and solar potential at building and urban scales, notably for parameter design and optimization analysis. Thus, the Ladybug and Honeybee plugins of the Ladybug Tools are employed in this research to conduct simulation and analysis. The study initiates parameter analysis by randomly altering the community's morphology to model energy consumption and solar PV generation, utilizing environmental performance simulation engines such as Radiance and OpenStudio. Subsequently, the Wallacei Genetic Optimization Plugin is used to conduct genetic operations to determine the most effective schemes.

2.3.2. Optimization Method

Genetic algorithms are well-suited for optimizing community morphology due to their fundamental principle, which involves utilizing continuous iterative processes, incorporating crossover and mutation mechanisms to favor individuals with higher fitness levels as prospective parents, ultimately converging on the individual with the highest fitness as the optimal solution. This philosophy aligns with the core premise of community morphology optimization, which iteratively combines morphological factors to pinpoint the individual exhibiting the highest fitness level. Optimization techniques rooted in genetic algorithms have gained widespread application, with the Wallacei plugin serving as a noteworthy example utilized for investigating optimization designs across building and urban contexts. Specifically, the Wallacei plugin is used to explore architectural forms adapted to solar radiation [43] and algorithms for the structure of urban morphology [44,45]. By employing a stochastic global search optimization method based on the genetic algorithm, the Wallacei plugin within the Grasshopper parametric platform is used to optimize the community morphology in this study. Functioning akin to the replication, crossover, and mutation of chromosomal genes during biological evolution, the algorithm transcends optimal solution determination by treating variable parameters as chromosomal genes, where the closeness of the optimization outcome to the target goal represents the individual's fitness level. Consequently, the optimization process adheres to the evolutionary principle of "survival of the fittest", yielding refined results aligned with one or multiple objectives. The parameters of the genetic algorithm are summarized in Table 3. The study takes morphology factors as variable parameters and reduces energy consumption, increases solar PV potential, and increases energy self-sufficiency as optimization objectives to perform optimization operations on the community morphology. Since the differences and uncertainty of the households' energy use behaviors are not considered, it is assumed that surplus power can be stored without limit; thus, the calculation of energy self-sufficiency is as follows:

$$\eta = E_S / E_D$$

where $\eta$ is the energy self-sufficiency rate, $E_S$ is energy produced on-site, and $E_D$ is energy use. When $\eta$ is 1, zero-energy performance is achieved.

**Table 3.** Parameter settings of the genetic algorithm.

| Parameters of the Genetic Algorithm | Value of the Parameters |
| --- | --- |
| Generation Size | 50 |
| Generation Count | 100 |
| Crossover Probability | 0.9 |
| Mutation Probability | 0.1 |
| Crossover Distribution Index | 20 |
| Mutation Distribution Index | 20 |

### 2.3.3. Optimization Process

In the optimization process, building location, building orientation (community orientation), and number of floors are considered variables. In contrast, community dimensions, floor area ratio, building density, and building type are treated as invariants. Building energy consumption and solar PV generation are the dependent variables in this study. The parameter settings are translated into Grasshopper parameterized language, and a genetic algorithm is utilized to execute iterative operations on the community morphology. The process consists of six steps, and the flow chart is presented in Figure 3.

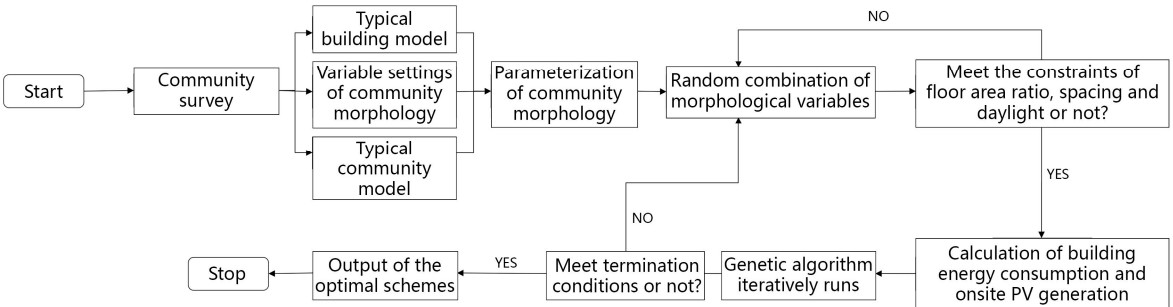

**Figure 3.** The flowchart of the optimized process.

In the first step, the center point of the community is used as the datum point for its orientation, and by default, all dwellings on the site are oriented in the same direction as the community, and the orientation of the community and the dwellings changes with the rotation angle of the datum point. The variation gradient is set to be 15°, and the variation range is from −45° to 45°. The rotation is counterclockwise from the positive north–south direction, which is taken as 0°.

In the second step, 437 grids are created by dividing the site area into orthogonal networks, in line with previous research [46], where each cell grid measures 7 m × 7 m. The center point of each grid serves as the building's datum point, allowing us to track variations in building location through changes in the datum point. The optimization process can become time-consuming as the number of buildings increases due to the various combinations of building positions that must be considered. To enhance optimization efficiency, we fixed the positions of the three dwellings in the northernmost row while allowing the other four dwellings to vary their positions randomly (Figure 4).

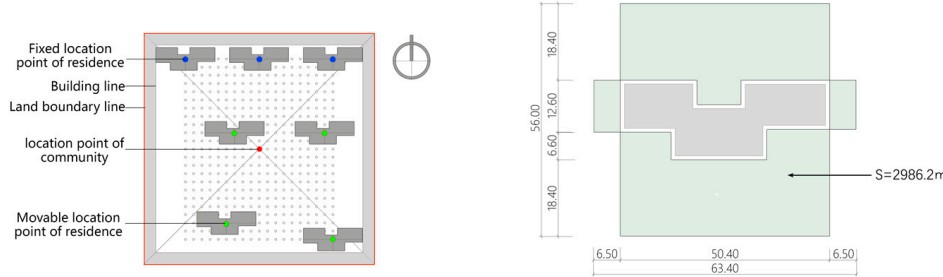

**Figure 4.** Locating the buildings and the constraints of building spacing.

In the third step, the seven dwellings are set with the information of layer height and number of floors, respectively. By referring to the standard for urban residential area planning and design [47], Project Code for Residential Buildings (draft for public comment) [48], and regulations on height limits for residential buildings inside and outside Hunan province, the floor height of the dwellings is set to be 3 m, and the number of floors varies from 19 to 26 floors. The morphology variables, as above, are summarized in Table 4.

**Table 4.** Community morphology variable settings.

| Community Morphology Variables | Values of Variable Parameters |
|---|---|
| Building location | Grid size: 7 m × 7 m<br>Grid center points: 473 |
| Building orientation | −45°, −30°, −15°, 0°, 15°, 30°, 45° |
| Number of floors | 19~26 floors |

In the fourth step, a parameterized platform is used to implement random combinations of the morphology variables, with restrictions on daylight hours and building spacing according to the Code for Technical Management of Urban and Rural Planning in Hunan Province (Trial) [49] and GB 50016-2014 Code for Fire Protection Design of Buildings (2018 Edition) [50], and with the full-window daylight hours for dwellings fulfilling that the minimum duration of adequate sunshine on a cold day is 2 h. Moreover, the building spacing is also restricted by setting the sum of the peripheral area and base area of individual dwellings. According to the calculation, the requirement of building spacing can be met when the sum of the peripheral area and base area of each dwelling is equal to 20,903.4 m$^2$ (Figure 4).

In the fifth step, dynamic simulation is carried out under the parameter design and the restriction settings above to calculate the associated building energy consumption and solar PV generation.

In the last step, the genetic algorithm is used to carry out iterative operations on the community morphology. The operation parameters are presented in Table 3. The optimization termination conditions are met when the iterative operation tends to converge and approaches stability. As a result, the final outputs of the optimization are those achieving the optimal objectives of low energy consumption, high PV potential, and high energy self-sufficiency.

## 3. Results

The Windows 11 system (Intel I9-13900k, 24 cores, 128 GB of RAM, 3.00 GHz) was subjected to 5000 iterative operations, lasting about 420 h. The genetic algorithm's iterations are determined by the population size (50) and the number of populations (100). Throughout the iterations, the optimization objective values gradually converged and stabilized. For ease of analysis and comparison, the predicted building energy consumption and electricity generated by PV are measured by kWh per unit of floor area, which can be obtained by dividing the total building energy consumption and electricity generation by the entire building floor area. As shown in Figure 5, the average energy consumption per unit of floor area decreased significantly at the start, followed by a slower decline to a minimum, ultimately stabilizing around 21.4 kWh/m$^2$, indicating a 4.26% reduction rate; the average electricity generated by solar PV per unit floor area increases gradually from 5.8 kwh/m$^2$ to 7.9 kwh/m$^2$ with the advancement of optimization and ends with maintaining at about 8.3 kwh/m$^2$; the energy self-sufficiency (i.e., the ratio of total solar PV power generation to whole-building energy consumption) presents a trend similar to that of solar PV generation, such as increasing gradually from 26% to 37% and stabilizing eventually at 39.1%. It implies that the varying morphology has a limited impact on building energy consumption. As a result, the optimization of energy self-sufficiency is dominated by the maximization of solar PV generation. However, 100% energy self-sufficiency cannot be achieved, even in the best case.

As above, 5000 schemes and datasets were generated during the optimization process. After removing duplicates and invalid entries, 205 viable schemes were identified. To quantify community morphology, measurable parameters such as building nearest neighbor index, community enclosure, maximum building scattered, and minimum building scattered were incorporated for the result analysis. Table A3 shows how the morphological parameters are calculated. Subsequently, a detailed analysis was performed on the qualitative and quantitative relationships between morphological parameters and build-

ing energy consumption, electricity generated by solar PV, and energy self-sufficiency. This analysis involved (1) taking building energy consumption, solar PV generation, and energy self-sufficiency rate as the dependent variables to identify morphological parameters of significance and explore the fitting relationship between the dependent variables and the morphological parameters of significance; and (2) analyzing the characteristics of the morphological parameters towards the optimal objectives before translating them into strategies.

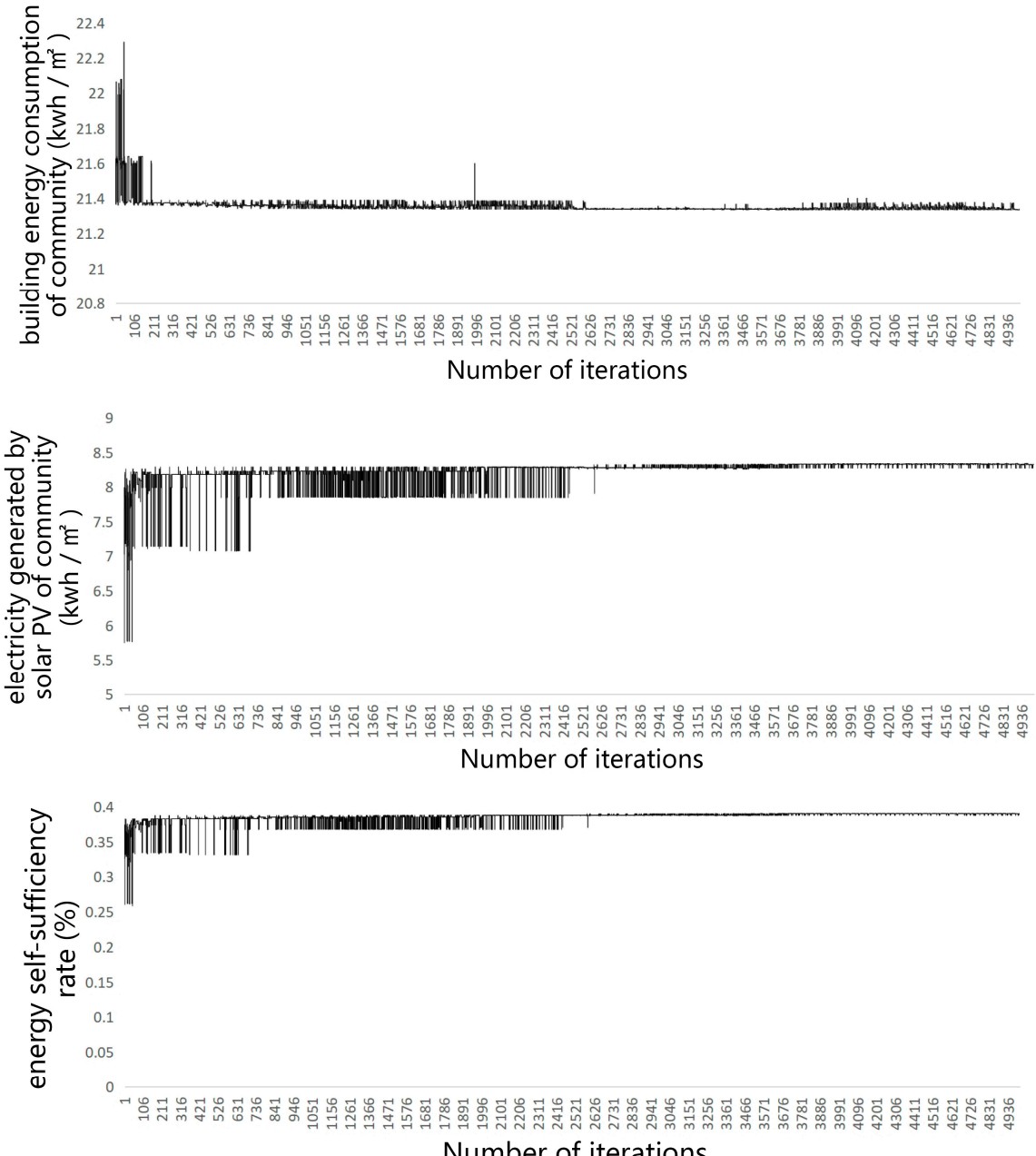

**Figure 5.** Iterative line graph of building energy consumption (**top**), electricity generated by solar PV (**middle**), and energy self-sufficiency (**bottom**).

### 3.1. Correlation Analysis

In this stage, SPSS is utilized to analyze the correlation between various objective values, such as building energy consumption, electricity generated by solar PV, energy self-sufficiency, and morphological parameters. These parameters include community orientation, building nearest neighbor index, community enclosure degree, maximum

building scattered degree, and minimum building scattered degree. The correlation analysis comprises two main components: A significance analysis aimed at identifying the morphological parameters that are statistically significant and a fitting analysis intended to confirm the parameter ranges that have a positive effect.

### 3.1.1. Significance Analysis

Table 5 presents the results of the significance analysis, where Pearson's index is employed to evaluate the impact of morphological parameters on building energy consumption, electricity generated by solar PV, and energy self-sufficiency. Building energy consumption exhibits positive correlations with all parameters, with the strongest correlation observed with community orientation. Notably, the correlation with the maximum building scattered degree is more pronounced than the minimum one. In contrast, the association with community enclosure degree is relatively weaker, with the building nearest neighbor index being the least pronounced. Conversely, electricity generated by solar PV is negatively correlated with all morphological parameters: the strongest correlation is with community orientation, with a Pearson's index even higher than those of building energy consumption and community orientation; it is then followed by the maximum building scattered degree and enclosure degree; its correlation with the minimum building scattered degree is weak, and the correlation with the building nearest neighbor index is the least significant. Regarding energy self-sufficiency, its correlation with the morphological parameters is largely convergent with solar PV generation. It should be noted that the correlation of energy self-sufficiency with community orientation is the most significant, while the impact of the building nearest neighbor index is not effective and thus will not be discussed in subsequent analysis. Since the morphological parameters are positively correlated with building energy consumption and negatively associated with solar PV generation and energy self-sufficiency, the effect of the morphological parameters on reducing building energy consumption is consistent with increasing solar PV generation and energy self-sufficiency.

**Table 5.** Correlation of building energy consumption, electricity generated by solar PV, and energy self-sufficiency with the morphology parameters.

| | Community Orientation | Building Nearest Neighbor Index | Enclosure Degree | Maximum Building Scattered Degree | Minimum Building Scattered Degree |
|---|---|---|---|---|---|
| building energy consumption (kWh/m$^2$) | 0.599 ** | 0.085 | 0.184 ** | 0.322 ** | 0.217 ** |
| electricity generated by solar PV (kWh) | −0.655 ** | −0.063 | −0.216 ** | −0.271 ** | −0.165 * |
| energy self-sufficiency (kWh) | −0.659 ** | −0.068 | −0.219 ** | −0.282 ** | −0.173 * |

* and ** denote significant correlations at the 0.05 level (two-tailed) and 0.01 level (two-tailed).

### 3.1.2. Fitting Analysis

To further identify the parameter ranges with a positive impact on building energy consumption, solar photovoltaic power generation, and energy self-sufficiency, a fitting analysis is conducted with the morphological parameters and the objectives. By comparing the fit degree of various regression equations, including linear, compound, logarithmic, quadratic, and cubic, it is determined that the cubic curvilinear regression equation offers the best fit, as depicted in Figure A5.

Among all, the R-square of the equation fitting the building energy consumption to the community orientation reaches 0.966, indicating that 96.7% of the variations in building energy usage can be attributed to changes in community orientation. a, e, and I in Figure A5 show that building energy consumption, electricity generated by solar PV, and energy self-sufficiency rate vary with some regularity as the community orientation

changes. However, the trends are not the same. For example, taking 0° (i.e., positive north–south direction) as the turning point, in the interval from −30° to 0°, as the orientation degree increases (counterclockwise rotation), the building energy consumption tends to decrease, and the amount of electricity generated by solar PV and the energy self-sufficiency rate gradually increase; in the interval from 0° to 45°, as the angle increases, the building energy consumption gradually increases, and the amount of electricity generated by solar PV and the energy self-sufficiency rate present a decreasing tendency. Therefore, the most conducive orientation to reduce building energy consumption and increase electricity generated by solar PV and energy self-sufficiency is 0°, i.e., north–south direction, followed by 15° west–south, 15° and 30° east–south, and lastly by 30° and 45° west–south, with the highest building energy consumption and the lowest electricity generated by solar PV and energy self-sufficiency. Additionally, the orientation of 45° east–south is excluded due to its failure to meet the required residential sunshine hours in the region.

b, c, d, g, h, f, j, k, and l in Figure A5 show that the fit degree of the objectives to community enclosure and building scattered degree are all low. Nevertheless, an analysis of the distribution of the scatters in the scatterplots shows that the enclosure degree varies from 0.25 to 0.33, the maximum building scattered degree varies from 3 to 9, and the minimum building scattered degree varies from 3 to 12. It can be seen that the objectives of minimum building energy consumption, maximum solar PV generation, and energy self-sufficiency rate are most likely to be achieved when the enclosure degree is in the low range, the maximum building scattered degree is in the high range, and the minimum building scattered degree is in the medium range.

This analysis underscores that community orientation presents the highest fit degree with building energy consumption, electricity generated by solar PV, and energy self-sufficiency rate, and a north–south-oriented community is most favorable for reducing building energy consumption, increasing electricity generated by solar PV, and maximizing energy self-sufficiency rate. In contrast, the enclosure degree and the scattered building degree display poor alignment with the objectives. Further analysis to explore the relationship between the parameters and optimization objectives is essential to identifying the most suitable optimization strategies. Thus, the following analytical step will involve screening the valid results to pinpoint optimal schemes for each optimization objective.

### *3.2. Optimization Strategies*

The optimal schemes towards individual optimization objectives are filtered out from the 205 valid results, including 27 schemes with the lowest building energy consumption, 23 with the highest solar PV generation, and 26 with the maximum energy self-sufficiency. Statistical analysis is then carried out regarding the characteristics of their morphological parameters and community layouts.

### 3.2.1. The Optimal Morphological Parameters

The analysis of the morphological parameters of the optimal schemes is presented in Tables A4–A6. The correlation characteristics between community orientation and objectives align with the conclusions drawn in the fitting analysis of "Section 3.1.2"; therefore, they will not be further discussed. Optimal parameter values for reducing building energy consumption include 0.25 or 0.28 for the enclosure degree, 6 or 9 for the maximum building scattered degree, and 9 or 12 for the minimum building scattered degree. When enhancing solar PV generation and energy self-sufficiency, the optimal parameter values are 0.25 for the enclosure degree, 9 for the maximum building scattered degree, and 9 or 12 for the minimum building scattered degree.

As mentioned above, there is no doubt that the selection of an appropriate community orientation is conducive to significantly reducing building energy consumption, increasing solar PV generation, and promoting energy self-sufficiency. The coincidence of the optimal parameter values towards improving solar PV generation and energy self-sufficiency confirms that the maximization of energy self-sufficiency is dominated by the increase of

on-site solar PV generation. Through fitting analysis, it becomes apparent that minimizing building energy consumption is attainable when the enclosure degree falls within the low or middle range, the maximum building scattered degree is in the middle or high range, and the minimum building scattered degree is in the high range. It implies that a layout featuring a horizontally open and vertically low-scattered design may not effectively reduce building energy consumption. Conversely, maximum solar PV generation and energy self-sufficiency can be realized when the enclosure degree is at the valley value, the maximum building scattered degree is at the peak value, and the minimum building scattered degree is in the high range. It implies that a layout characterized by close horizontal proximity and vertical scattering is advantageous for increasing on-site PV generation and community energy self-sufficiency. Furthermore, to translate these optimal parameter values into actionable strategies and account for any potential oversights of other contributing factors, an in-depth examination of the layout characteristics of the optimal schemes is conducted in the subsequent stage.

### 3.2.2. The Optimal Community Layouts

The layout characteristics of the optimal schemes displayed in Tables A4–A6 are further analyzed in Table A7. The schematic diagram illustrating the division of the layout direction is provided in Figure 6. It reveals that, in terms of vertical layout, most optimal schemes follow a consistent pattern toward all three objectives. This pattern includes positioning high in the south and low in the north for both the first and middle columns, positioning low in the south and high in the north for the end column, and maintaining a flush position for the first row. Moreover, the middle row is high in the west and low in the east, and the end row is low in the west and high in the east. Regarding the horizontal layout, the optimal schemes towards minimum energy consumption and maximum energy self-sufficiency adopt staggered layouts. On the other hand, the majority of the optimal schemes aim for maximum solar PV generation and feature row layouts for the first row and staggered layouts for the middle row.

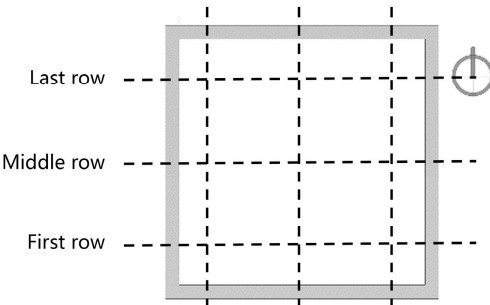

**Figure 6.** Diagram of the division of the layout pattern.

The optimal schemes toward all objectives demonstrate a consistent pattern of horizontal and vertical layouts despite variations in morphological parameters. This uniformity provides an effective and universal layout pattern for the design and planning of zero-energy communities. Notably, a horizontally staggered layout, in addition to being horizontally close and vertically scattered, is particularly advantageous for maximizing energy self-sufficiency. As it is known that a horizontally staggered layout is conducive to site ventilation, the ventilation environment can be improved along with energy self-sufficiency in this case. Furthermore, an analysis of vertical layout reveals that the traditional practice of low south and high north may not be wholly suitable for zero-energy communities. These findings are further detailed in Table 6 to summarize the layout characteristics essential for sustainable community planning.

**Table 6.** Characteristics of the layouts of the optimal schemes.

| Optimization Objective | Optimizing Trend | Vertical Layout (Building Height) | | | | | Horizontal Layout | | |
| | | North–South | | East–West | | | North–South | East–West | |
| | | First and Middle Column | Last Column | First Row | Middle Row | Last Row | First, Middle and Last Column | First Row | Middle Row |
| Building energy consumption | Reduce | | | | | | | Staggered | |
| Electricity generated by solar PV | Increase | South > north | South < north | Flush | West > east | West < east | Staggered | Row | Staggered |
| Energy self-sufficiency | | | | | | | | Staggered | |

## 4. Discussion and Conclusions

As above, parameter design and automatic optimization are conducted using the morphological factors of a typical community model extracted from a survey of residential communities in Changsha, a typical HSCW city. The correlations between morphological parameters and optimization objectives are analyzed to derive optimization strategies to achieve energy self-sufficiency. A discussion of the results is presented below:

1. The optimization of community morphology can reduce building energy consumption by 4.26% and increase solar PV generation and energy self-sufficiency by 45% and 13.2%, respectively. It implies that although the optimization may have a modest effect on energy efficiency, it can significantly boost solar PV generation, ultimately leading to a substantial enhancement in community energy self-sufficiency, which is significant in realizing zero-energy communities. Notably, under the restriction conditions on the floor area ratio and the average number of floors, etc., as recommended by the current standards for residential developments, the maximum energy self-sufficiency rate that can be achieved is 39%.

2. Optimizing morphological parameters to reduce building energy consumption aligns with optimizing them to increase solar PV generation, suggesting a synergistic relationship in morphology optimization. In addition, the optimal parameter values for enhancing solar PV generation and energy self-sufficiency confirm that maximizing energy self-sufficiency depends on increasing on-site solar PV generation.

3. Among the morphological parameters, community orientation presents the strongest correlation with energy self-sufficiency, followed by the maximum building scattered degree and enclosure degree, then the minimum building scattered degree. Conversely, the correlation between the building nearest neighbor index and energy self-sufficiency is negligible. Therefore, it is essential to carefully select an appropriate community orientation, as it plays a crucial role in achieving energy self-sufficiency in community planning and design.

4. The optimal community orientation towards maximizing energy self-sufficiency is the north–south orientation, followed by 15° west–south, 15° and 30° east–south, and lastly by 30° and 45° west–south. For the other parameters, the maximum energy self-sufficiency can be achieved when the enclosure degree is at the valley value, the maximum building scattered degree is at the peak value, and the minimum building scattered degree is in the high range. These parameters collectively suggest that a horizontally close and vertically scattered layout is favorable for maximizing energy self-sufficiency.

5. The study determined a specific layout pattern best suited for the area. The favorable vertical layout is high in the south and low in the north for the first and middle columns; low in the south and high in the north for the last column; the same height for the first row; high in the west and low in the east for the middle row; and low in the west and high in the east for the last row. These findings suggest that the traditional vertical layout of low south and high north may not fully apply to zero-energy communities. Furthermore, the favorable horizontal layout is staggered. It should be noted that the study did not consider the overshadowing surrounding the site, so the conclusions obtained here may possess some limitations.

Since energy self-sufficiency cannot be achieved with only morphology optimization, in order to realize zero-energy communities in the future, it may be necessary to integrate the utilization of other renewable energy sources or even make fundamental adjustments to the current standards of land use to increase the amount of on-site PV generation per unit floor area. Additionally, it can be concluded that a vertically scattered, horizontally closed, and staggered layout is favorable for maximizing energy self-sufficiency. The exploration of the optimization effect and strategies of the community morphology in this study will provide a research basis for the promotion of low-energy communities and renewable energy-sharing communities in the area and prepare for the development of zero-energy communities in the future. Notably, in order to expedite the optimization process, the community morphology is partially defined by the horizontal position of buildings in the last row, and only the main dwelling type is considered. A more comprehensive study will be carried out by removing the limitation step by step with consideration for a mix of different dwelling types and by managing each objective's convergence and stabilization time to decrease computing time and enhance optimization efficiency. Above all, through optimizing community morphology, the study contributes to the reduction of community carbon emissions by reducing energy consumption, increasing on-site solar PV generation, and, as a result, maximizing energy self-sufficiency.

**Author Contributions:** Conceptualization, X.L.; methodology, X.L. and Y.Z.; software, Y.Z.; validation, Y.Z. and H.L.; formal analysis, Y.Z.; investigation, Y.Z., X.L. and H.L.; resources, H.L.; data curation, Y.Z.; writing—original draft preparation, Y.Z.; writing—review and editing, X.L. and X.X.; visualization, Y.Z. and X.X.; supervision, X.L. and H.L.; project administration, X.L.; funding acquisition, X.L. All authors have read and agreed to the published version of the manuscript.

**Funding:** The research presented in this paper was funded by the Department of Science and Technology of Hunan Province, China, through the Natural Science Foundation of Hunan (No. 2022JJ30140).

**Data Availability Statement:** Due to restrictions, the data presented in this study are available on request from the corresponding author.

**Conflicts of Interest:** The authors declare that they have no known competing financial interests or personal relationships that could have appeared to influence the work reported in this paper.

**Appendix A**

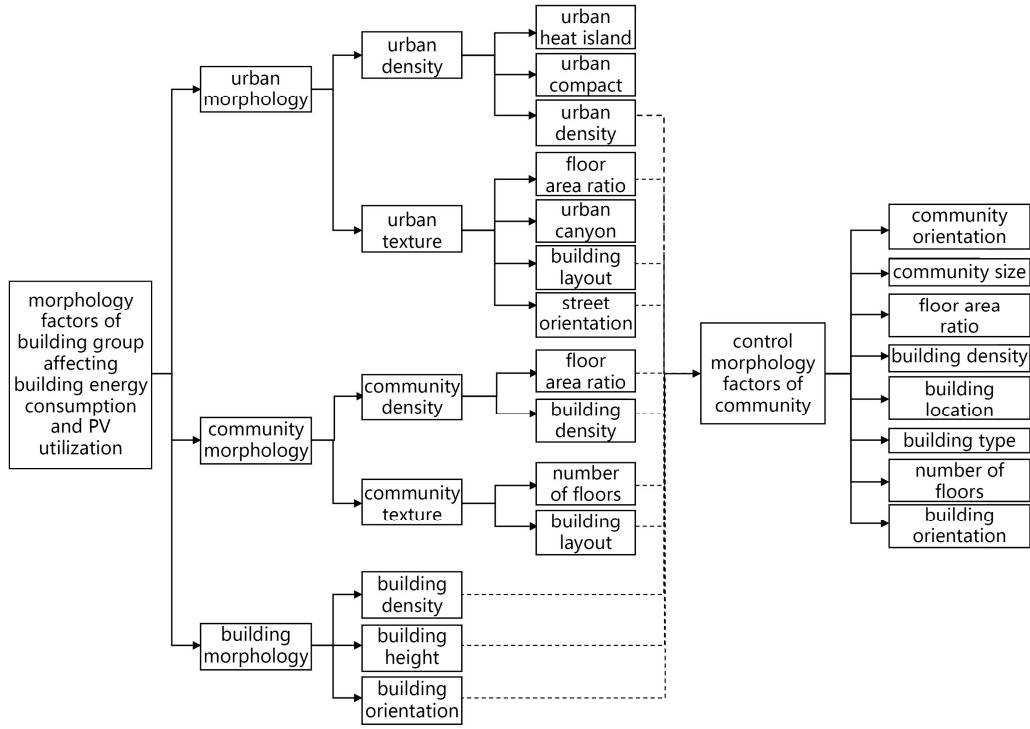

**Figure A1.** Overview of the morphological factors.

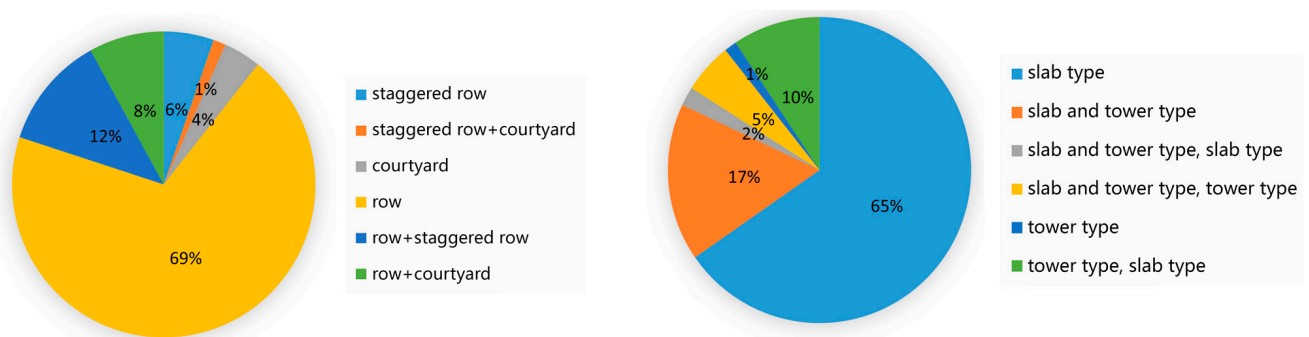

**Figure A2.** The composition of community layouts and building types.

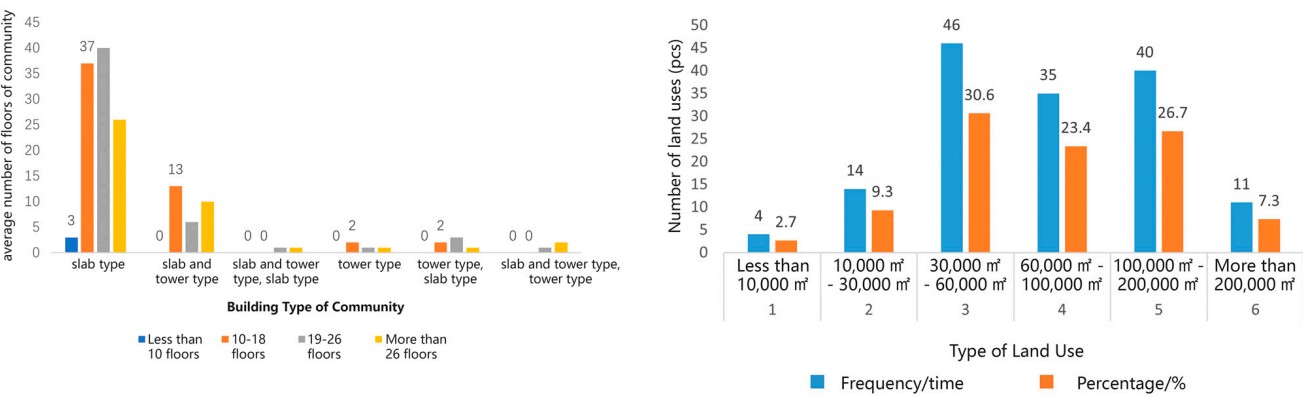

**Figure A3.** The frequency distribution of the communities in terms of the average number of floors and land area.



**Figure A4.** Row-type layout.

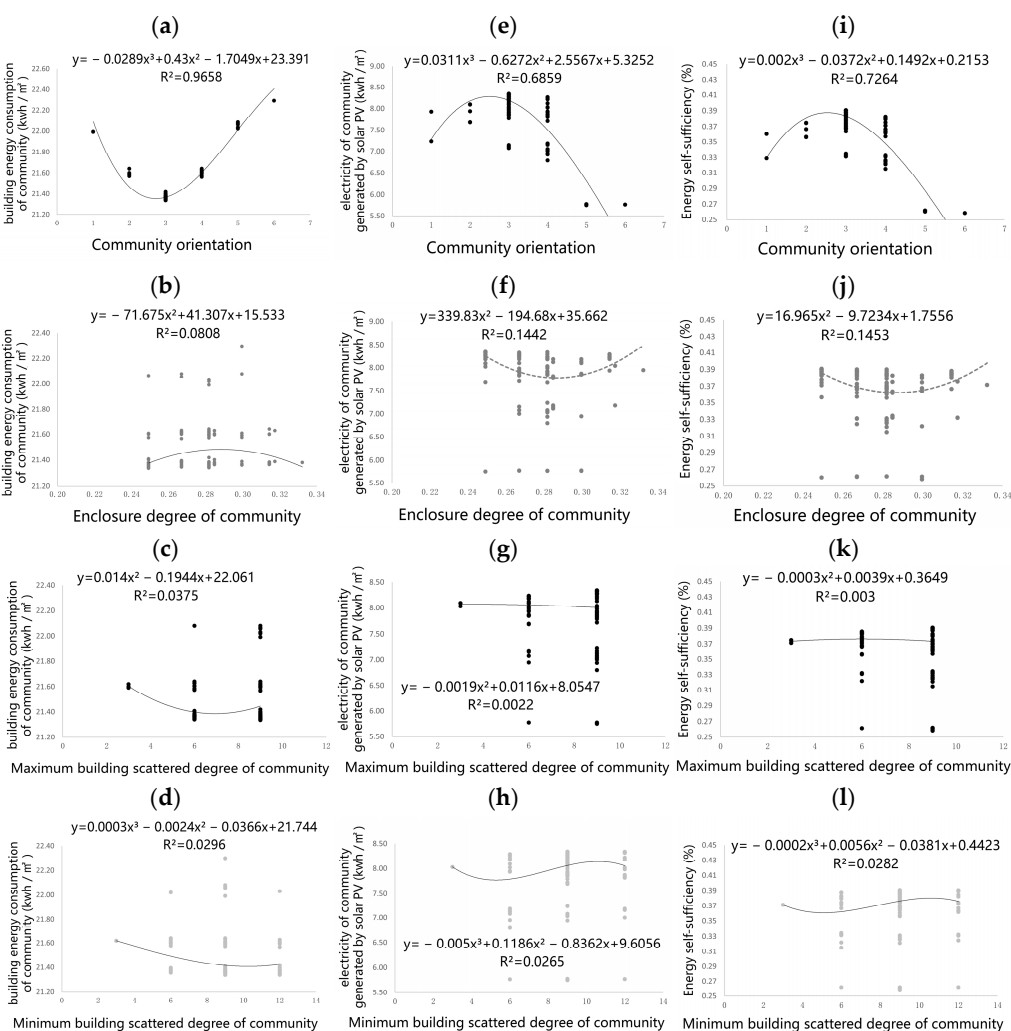

**Figure A5.** The fitting relationship between the objective values and the morphological parameters ((**a**)—between building energy consumption and orientation; (**b**)—between building energy consumption and enclosure degree; (**c**)—between building energy consumption and maximum building scattered degree; (**d**)—between building energy consumption and minimum building scattered degree; (**e**)—between electricity generated by solar PV and orientation; (**f**)—between electricity generated by solar PV and enclosure degree; (**g**)—between electricity generated by solar PV and maximum building scattered degree; (**h**)—between electricity generated by solar PV and minimum building scattered degree; (**i**)—between energy self-sufficiency and orientation; (**j**)—between energy self-sufficiency and enclosure degree; (**k**)—between energy self-sufficiency and maximum building scattered degree; (**l**)—between energy self-sufficiency and minimum building scattered degree).

## Appendix B

**Table A1.** Correlation studies of morphological factors of communities with building energy consumption and PV utilization.

| Year | Research Object | Climate | Impact Object | Morphological Factor | Software | Reference |
|---|---|---|---|---|---|---|
| 2013 | neighborhood | Temperate maritime climate | Heating, cooling, and embodied energy | Number of floors (high-rise, mid-rise, and low-rise) | Virvil and HTB2 | [18] |
| 2015 | residence | Hot Summer and Cold Winter zone | Mix | Building orientation | DesignBuilder | [23] |
| 2016 | neighborhood | Hot Summer and Cold Winter zone | Heating, cooling, etc. | Urban density | UMI Building energy modeling plugin | [19] |
| 2019 | community | Hot Summer and Cold Winter zone | Heating and cooling | Floor area ratio, shape factor, building density, average number of floors, building orientation, and standard deviation of building heights | HTB2 and Virvil | [21] |
| 2019 | neighborhood | Temperate maritime climate | Heating and cooling | Coverage rate and shape factor | CitySim and Meteonorm | [22] |
| 2020 | neighborhood | Subtropical Mediterranean climate | Heating and cooling | Combined layout of the building group | Grasshopper | [20] |
| 2019 | neighborhood | Temperate grassland climate | Solar access | Site layout of the neighborhood and associated position of buildings | Energyplus | [29] |
| 2022 | community | Cold region | Heating | Building spacing coefficients, vertical layout, and building group orientation | Design Builder | [25] |
| 2015 | community | Temperate maritime climate | Solar radiation | Floor area ratio, building average spacing, standard average of building heights, and average building perimeter | ArcGIS | [30] |
| 2017 | community | Temperate maritime climate | Solar PV generation potential | Building orientation, building height, and building spacing | Rhion and Radiance | [28] |
| 2019 | neighborhood | Tropical rainforest climate | Solar PV generation potential | Building and neighborhood types | Energyplus and Radiance | [27] |
| 2023 | community | Tropical region | Solar PV generation potential | Building density and spacing | ArcGIS | [31] |

**Table A2.** Parameter settings of the basic building model.

| Parameter | | Value |
|---|---|---|
| Window-to-wall ratio | North-facing | 0.3 |
| | South-facing | 0.3 |

**Table A2.** *Cont.*

| | Parameter | Value |
|---|---|---|
| Heat transfer coefficient of the building envelope (W·m$^{-2}$·K$^{-1}$) | Exterior wall | 1 |
| | Party wall | 1.5 |
| | Elevated floor where the underside is exposed to outdoor air | 1 |
| | Floor | 1.8 |
| | Roof | 0.4 |
| | Exterior window | 2.5 |
| Power output | Interior service power/(W·m$^{-2}$) | 3.8 |
| | Lighting power/(W·m$^{-2}$) | 5 |
| | Occupant density (ppl·m$^{-2}$) | 0.04 |
| | Air exchange rate per hour | 1 |
| Heating, ventilation, and air conditioning (HVAC) system | Minimum fresh air volume per capita/(m$^3$·h$^{-1}$) | 30 |
| | Heating temperature/°C | 18 |
| | Cooling temperature/°C | 26 |
| | COP (heating) | 2.9 |
| | COP (cooling) | 3.2 |
| | Period | Heating: 1 December–28 February Cooling: 15 June–31 August |

**Table A3.** A summary of the morphological parameters.

| Community Morphology Factor | Morphology Factor Parameter | Definition | Calculation Formula |
|---|---|---|---|
| Community orientation | Orientation | 0° in a positive north–south direction, negative in a clockwise direction, positive in a counterclockwise direction | N/A |
| Building location | Building nearest neighbor index (BNNI) | Describing the spatial distribution characteristics of point elements. The smaller the index, the more discrete the elements are, and conversely, the more clustered they are. It is equal to the ratio of the actual nearest neighbor distance to the theoretical nearest neighbor distance (random distribution). | $BNNI = \dfrac{\overline{D_O}}{\overline{D_E}}$ $\overline{D_O} = \dfrac{\sum_{i=1}^{n} d_i}{n}$ $\overline{D_E} = \dfrac{0.5}{\sqrt{\frac{n}{A_s}}}$ $d_i$ is the distance from the base center of the ith building to the base center of the building nearest to it. n is the number of buildings. $A_s$ is the land area of the community. |
| | Enclosure degree (ED) | Measuring the horizontal enclosure characteristics of the site. It is equal to the ratio of the outer perimeter of the building group within the site to the perimeter of the building control line. | $ED = \dfrac{\sum_{i=1}^{n} d_i}{m}$ $d_i$ is the length of the bottom edge of the ith outer façade. m is the perimeter of the building control line |
| Number of floors of buildings | Building scattered degree (SD) | Characterizing the vertical distribution of the dwellings in a community. The maximum building scattered degree (SD$_{MAX}$) is the difference between the maximum building height and the average building height, and the minimum building scattered degree (SD$_{Min}$) is the difference between the average and minimum building heights within the site. | $SD_{MAX} = h_{max} - h_a$ $SD_{Min} = h_a - h_{min}$ $h_{max}$ is the highest building height $h_{min}$ is the lowest building height $h_a$ is the average building height |

**Table A4.** Morphological parameters of the optimal schemes for reducing building energy consumption.

There are 27 schemes with the lowest building energy consumption (21.34 kWh/m²)

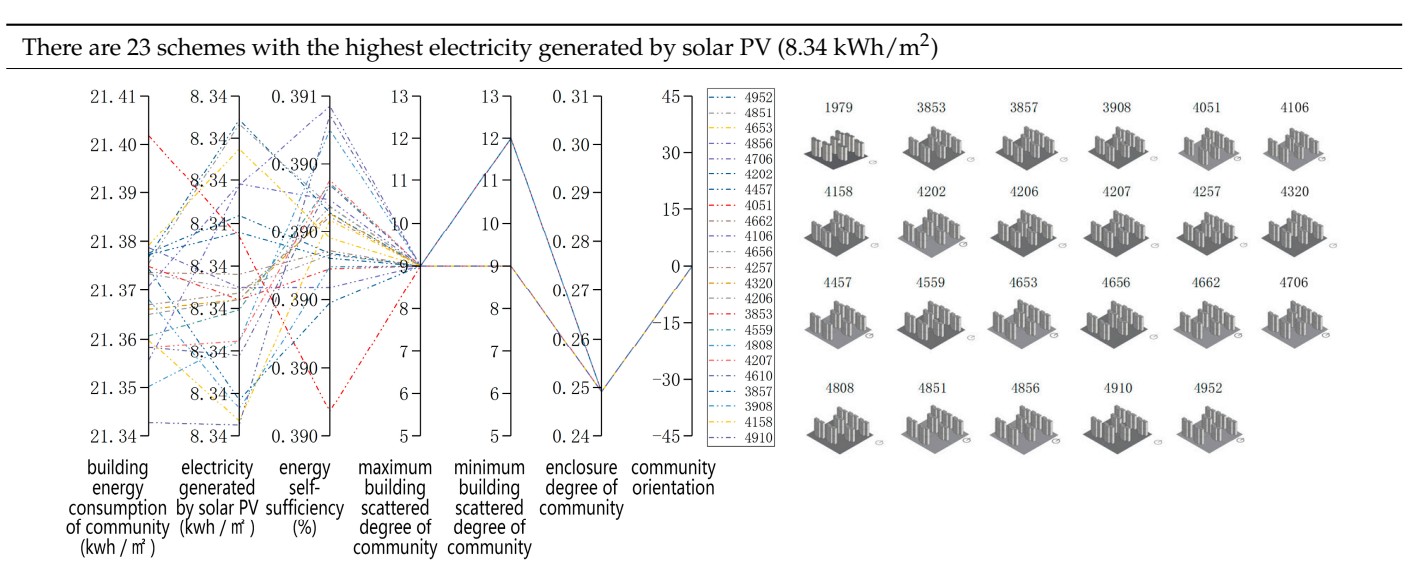

(1) Maximum building scattered degree

(2) Minimum building scattered degree

(3) Community enclosure degree

(4) Community orientation

**Table A5.** Morphological parameters of the optimal schemes for increasing solar PV generation.

There are 23 schemes with the highest electricity generated by solar PV (8.34 kWh/m²)

**Table A5.** *Cont.*

| (1) Maximum building scattered degree | (2) Minimum building scattered degree | (3) Community enclosure degree | (4) Community orientation |
|---|---|---|---|
| maximum building scattered degree of community | minimum building scattered degree of community | enclosure degree of community | community orientation |

**Table A6.** Morphological parameters of the optimal schemes for maximizing energy self-sufficiency.

There are 26 schemes with the highest energy self-sufficiency rate (39.0–39.1%)

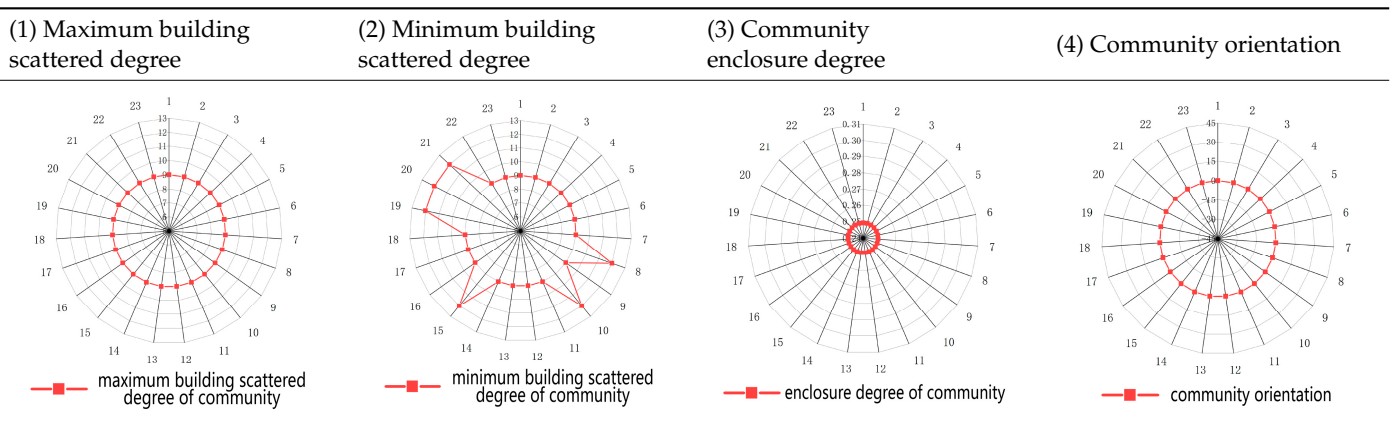

| (1) Maximum building scattered degree | (2) Minimum building scattered degree | (3) Community enclosure degree | (4) Community orientation |
|---|---|---|---|
| maximum building scattered degree of community | minimum building scattered degree of community | enclosure degree of community | community orientation |

**Table A7.** Statistics of the vertical and horizontal layouts of the optimal schemes.

| Type | Layout Direction | | | Objective 1: Minimum Building Energy Consumption | Objective 2: Maximum Solar PV Generation | Objective 3: Maximum Energy Self-Sufficiency |
|---|---|---|---|---|---|---|
| Vertical layout (building height) | North–south | First column | South > north | 100% | 100% | 100% |
| | | Middle column | South > north | 77.8% | 100% | 100% |
| | | | South > middle < north | 22.2% | | |
| | | Last column | South > north | 3.7% | | 4.3% |
| | | | South < north | 81.5% | 82.6% | 87.0% |
| | | | Flush | 14.8% | 17.4% | 8.7% |
| | East–west | First row | West < east | 44.4% | | |
| | | | Flush | 55.6% | 100% | 100% |
| | | Mid row | West > east | 100% | 100% | 100% |
| | | Last row | West < east | 100% | 100% | 100% |
| Horizontal layout (relative position) | North–south | First column | Staggered | 100% | 100% | 100% |
| | | Middle column | Staggered | 100% | 100% | 100% |
| | | Last column | Staggered | 100% | 100% | 100% |
| | East–west | First row | Row | | 14 (60.9%) | 9 (39.1%) |
| | | | Staggered | 27 (100%) | 9 (39.1%) | 14 (60.9%) |
| | | Mid row | Staggered | 27 (100%) | 23 (100%) | 23 (100%) |

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
