# Peer review of "Morphology Optimization of Residential Communities towards Maximizing Energy Self-Sufficiency in the Hot Summer Cold Winter Climate Zone of China"

_land, doi:10.3390/land13030337_

Round 1
Reviewer 1 Report
Comments and Suggestions for Authors
accept in present form
1. What is the main question addressed by the research?
Morphology optimization of high-rise residential communities
2. Do you consider the topic original or relevant in the field? Does it
address a specific gap in the field?
yes we face this problem in several fields
3. What does it add to the subject area compared with other published
material?
morphological indicators of 150 communities in Changsha, to establish a prototype model by Grasshopper platform
4. What specific improvements should the authors consider regarding the
methodology? What further controls should be considered?
...an urban design perspective...
5. Are the conclusions consistent with the evidence and arguments
presented and do they address the main question posed?
yes
6. Are the references appropriate?
references are appropriate for the specific topic
7. Please include any additional comments on the tables and figures.
----
Author Response
Please see the attachment(Cover letter+Author's Reply)

Reviewer 2 Report
Comments and Suggestions for Authors
The article shows a common theme addressed in numerous publications, regarding considering the morphology and orientations of buildings to know the levels of energy consumption or savings, both for heating and cooling; However, in this case the main interest is to evaluate other variables such as energy self-sufficiency and photovoltaic solar energy generation and their correlations, which is what generates interest in its reading.
Author Response

(The authors gave the same response as above.)

Reviewer 3 Report
Comments and Suggestions for Authors
The proposed research on Morphology optimization of high-rise residential communities towards maximizing energy self-sufficiency is highly interesting and provides sufficient evidence. It has the potential to greatly contribute to carbon reduction efforts. I have a few suggestions for the authors to consider during the minor revision:
Genetic algorithms are widely used for optimization purposes. It would be valuable for the authors to explain the advantages of genetic algorithms and provide a rationale for choosing this method for automating the optimization process. Additionally, including the optimization model's description in the form of equations or flowcharts would further elucidate the approach and improve clarity.
Furthermore, it is essential to include a detailed discussion on the limitations of the research. This section should address any constraints, assumptions, or potential drawbacks of the proposed methodology. Additionally, the authors should propose future approaches to reduce the time required for iterative operations and elaborate on how this work can specifically contribute to carbon reduction efforts.
By addressing these suggestions in the minor revision, the authors can enhance the overall quality and clarity of their research, making it more accessible and impactful to the readers.
Author Response

(The authors gave the same response as above.)

Reviewer 4 Report
Comments and Suggestions for Authors
The authors employ the Grasshopper platform to model the energy behavior of building communities in the Hot Summer and Cold Winter climate zone (HSCW) of China, based on the morphological indicators of 150 communities in Changsha. The community morphology is simulated and optimized with building location, orientation and number of floors as independent variables, and building energy consumption, PV generation and energy self-sufficiency rate as dependent variables. The results predict that morphology optimization could reduce building energy use by 4.26%, increase PV generation by 45%, and improve energy self-sufficiency by 13.2%. Of course, optimal self-sufficiency can reach 39%, if other parameters are modified in addition to morphology improvement. Community orientation is found to be the most significant morphological factor towards maximizing energy self-sufficiency. The south-north orientation results to the optimal behavior. This is an interesting work, useful to the architect - urban designer, however, its presentation will need very significant improvement in order to become readable.
The authors should address the following issues in a revised version of their manuscript:
The structure of the manuscript at the end of the Introduction section should comprise also the chapters/ sections involved.
The authors mention multiple cited papers, without specific comment for each one. This is not allowed in a scientific paper. Each cited paper must be recited separately with its own comments.
Line 48: Please define floor area ratio (FAR) in this instance.
Lines 76, 78, 81. Here you use a mixed citation style, adding the year of the publication. This is not allowed. Please use the style prescribed by this journal.
Figure 1 increase font size and figure resolution
Line 97 please define canyon aspect ratio, street aspect ratio, and space open rate.
Line 162 please define and use a better terminology for the “determinant” type
Table 3: some internal grid lines must be added to determine which entries of the second and third column correspond to each entry of the first column e.g. combine window-to-wall ratio with the next two entries of columns 2-3 etc
Figure 8 – main dimensions?
Lines 222-223 what you mean by “future”?
Line 322 you mean 5000 iterations? Please explain in more detail.
The quality of English is low. A thorough revision and improvement of the manuscript, preferably by a native speaker, will be necessary.
Comments on the Quality of English LanguageA thorough English revision and improvement of the manuscript, preferably by a native speaker, will be necessary.
Author Response

(The authors gave the same response as above.)
